# Impacts of Land-Use Changes on Vegetation and Ecosystem Functioning: Old-Field Secondary Succession

**DOI:** 10.3390/plants10050990

**Published:** 2021-05-16

**Authors:** Javier Pérez-Hernández, Rosario G. Gavilán

**Affiliations:** Botany Unit, Department of Pharmacology, Pharmacognosy and Botany, Faculty of Pharmacy, Complutense University, E-28040 Madrid, Spain

**Keywords:** abiotic filtering, biotic limit, chronosequence, dynamic, grasslands, local species pool, plant functional types traits, priority effects, regional species pool, review, species coexistence, soils traits, soil seed bank

## Abstract

The study of ecological succession to determine how plant communities re-assemble after a natural or anthropogenic disturbance has always been an important topic in ecology. The understanding of these processes forms part of the new theories of community assembly and species coexistence, and is attracting attention in a context of expanding human impacts. Specifically, new successional studies provide answers to different mechanisms of community assemblage, and aim to define the importance of deterministic or stochastic processes in the succession dynamic. Biotic limits, which depend directly on biodiversity (i.e., species competition), and abiotic filtering, which depends on the environment, become particularly important when they are exceeded, making the succession process more complicated to reach the previous disturbance stage. Plant functional traits (PFTs) are used in secondary succession studies to establish differences between abandonment stages or to compare types of vegetation or flora, and are more closely related to the functioning of plant communities. Dispersal limitation is a PFT considered an important process from a stochastic point of view because it is related to the establishing of plants. Related to it the soil seed bank plays an important role in secondary succession because it is essential for ecosystem functioning. Soil compounds and microbial community are important variables to take into account when studying any succession stage. Chronosequence is the best way to study the whole process at different time scales. Finally, our objective in this review is to show how past studies and new insights are being incorporated into the basis of classic succession. To further explore this subject we have chosen old-field recovery as an example of how a number of different plant communities, including annual and perennial grasslands and shrublands, play an important role in secondary succession.

## 1. Introduction

Ecological succession is a process whereby plant communities re-assemble after a natural or anthropogenic perturbation. Odum [1] formulated the secondary succession theory and further extended it from more specific studies of various ecological characteristics. Following Odum, ecological succession can be defined by three parameters: (i) it is an orderly process of plant community change that is directional enough to be considered predictable; (ii) it is the result of the modification of the physical context by the target plant community; and (iii) it ultimately achieves a stable ecosystem that maintains a maximum of biomass and mutualistic relationships between the organisms through the available unit of energy flux [1,2,3].

Until the last quarter of the 19th century, the forests of Western Europe were subjected to similar uses in both the north and south; however, northern countries subsequently recognized the importance of restoration, leading to the implementation of reforestation campaigns. At the end of the World War I, large farms began to appear in northern Europe resulting from the concentration of smaller ones [4]. Agricultural practices became mechanized, favoring the abandonment of lands and recovered forest. By the late 1960s and again in the 1990s, these phenomena were exacerbated by some of the EU’s agricultural policies which led to population migration and a recolonization of abandoned areas by natural vegetation [4,5,6]. Forests in southern Europe represented an important resource, whose destruction occurred without any opportunity for regeneration. They have been used for fuel or agriculture, in some cases maintaining a multi-purpose sustainable agroforestry system consisting of a mosaic of widely-spaced scattered oaks, known as a dehesa in Spain, for instance [4,7,8,9]. Secondary succession was not known or acknowledged in arid southern areas [10].

The Common Agricultural Policies (CAP) introduced by the European Union in 1990 reorganized lands to be more competitive for the global market. Several regions shifted to cultivating other plants owing to the cost of maintaining the old ones, in some cases due to lack of accessibility to cultivated areas. From the beginning of the CAP (1990–2010), 144,733 km^2^ of lands transitioned to grasslands and forest, with an increase of 150% compared to the 1970–1990 period [11]. This abandonment, together with the globalization process, also produced an increase in population migration due to changes caused by the inability of more traditional low-productivity agriculture to compete with more productive mechanized agriculture [12].

Land abandonment has opened up opportunities to evaluate the way vegetation recovers (Table 1). The practice of maintaining regional inventories on agriculture and forests began in the mid-19th century. Long-term studies were conducted on changes in vegetation cover and land use, some of which were limited to particular areas of France. They included secondary succession on grasslands and parcel vegetation mapping in France [13,14], but also in landscape since prehistory [15]; and studies on the development of lands in England (rivers, coastal areas, agricultural lands, etc.), long-term changes due to natural or anthropic changes [16].

Three main drivers have been described as being responsible for land abandonment [17]: (i) ecological drivers: certain environmental features of the land such as elevation, geological substrate, slope, soil depth, erosion and global change were important factors in the abandonment [18,19]; (ii) socioeconomic drivers, new economic opportunities, migration, rural depopulation, market incentives, etc., including the European Union‘s Common Agricultural Policies (CAP) [19,20,21]; and (iii) mismanagement, with practices such as over-exploitation and desertification [18,20].

The development of aerial photography in the 1940s created opportunities for studying changes in vegetation. In spite of the difficulties in distinguishing natural vegetation from agricultural land due to the poor quality of the pictures and the lack of vegetation studies, the comparison through aerial photographs confirms the increase in forest cover in the northern Mediterranean Basin in the second half of the 20th century [22,23].

Plant community functioning and succession patterns have been extensively studied and discussed (Table 1). Huston and Smith [24] established certain rules for the study of succession through the analysis of different models: (i) Lotka-Volterra, based on direct interactions in competition between individuals [25,26,27,28,29]; (ii) based on individuals and competition for resources [30,31,32]; (iii) competition for a wide range of resources [33,34,35,36,37,38,39]; and (iv) multispecies [2,24]. These rules are: Competition between individuals for resources exists in all plant communities. No significant differences have been found between inter-and intraspecific competition, and the processes of facilitation, tolerance or inhibition are relative and even simultaneous and have different weights during each successional stage.Plants can alter their environment, changing the availability of resources. General uniformity is higher in the intermediate stages of succession when the periods of coexistence are higher due to the dominance of slow-growing plants.Physiological and energy restrictions hinder species’ capacity to maximize their competitive character in any circumstance. This produces an inverse correlation between certain groups of traits and the relative competitive abilities of species change depending on a range of environmental conditions. These particular traits are different depending on whether the succession is in a late or early stage.

However, the dynamic complexity of plant interaction has recently been studied in depth, using approaches based on valid empirical inquiries which can test general patterns related to coexistence, detect higher rates of population growth at low abundances, validate how these mechanisms operate in the field, identify certain properties of the species that influence features (fitness) in these interactions, and test predictions in community structure models [24,40,41,42,43]). Species dominance and succession dynamism have been studied through different hypotheses [44]. The most outstanding finding is nutrient colonization-competition, which explains herb initial dominance (0–40 years). However, the nutrients vs. light competition hypothesis better accounts for the dominance of woody plants, they compete better in N rich fields displacing herbs. Both ideas predict the vegetation composition and successional dynamic through simple interspecific interactions [44].

## 2. Secondary Succession Dynamic: Deterministic or Stochastic Processes?

A secondary succession is a cluster of diverse processes affected by many factors, so the results may differ in each case. Understanding the dynamics of secondary succession will help us to better understand both the final and intermediate stages. There are two different frameworks to evaluate the establishment of plant communities: deterministic and stochastic. In the deterministic framework the local community dynamics are determined by specific species traits and local abiotic or biotic factors. In the stochastic view, the dynamics are determined only by the demographic stochasticity and dispersal limitation. In the deterministic model the beta-diversity decreases along the succession (heterogeneity or difference in local diversity between communities). In the stochastic model, the divergent species composition becomes greater as the beta-diversity increases along the succession [45,46,47]. There is some debate around these concepts, materialized in the ecological niche theory and the neutral theory of biodiversity [48,49,50]. The first case follows a more deterministic idea: differences between niches are based on soil, climate resources and competitors. The neutral theory of biodiversity follows the stochastic idea based on the dispersal process or random extinctions of organisms. Some succession studies support this idea, i.e., in subtropical forests stochastic processes such as functional changes, species richness or phylogenetics were dominant [47]. However, both aspects play an important role in structuring plant communities and consequently in the succession process [51,52,53].

The succession pattern and its speed depends on the species pool, their biology, local ecological conditions, and the landscape surrounding the abandoned land [22,48,54,55]. For instance, on abandoned land, an intermediate, degraded stage could have the appearance of a final stage, although it is actually the legacy of cultivation that is acting, masking an intermediate stage that may be longer than expected [22]. There is little knowledge of uninterrupted succession processes in Mediterranean areas due to the presence of grasslands in intermediate stages that have undergone frequent fires that alter the re-naturalization process. The absence of woody species in the surroundings of abandoned land can lead to increased erosion processes and the frequent collapse of the succession [56].

Both the deterministic and the stochastic frameworks bring information on the dynamic of secondary succession and are necessary to explain the processes that occur on it. In Mediterranean climates, more studies on secondary succession are necessary to have a better look on the processes.

### Passive and Spontaneous Regeneration

The colonization of abandoned lands by passive and spontaneous regeneration can occur through any type of species or animals that can establish, survive, and grow there. It is usually a quick process in very productive areas such as tropical or humid zones but very slow in environments with low primary production such as the Mediterranean [17]. Succession in abandoned, cultivated areas may be rapid with only a transition stage of grasslands, quickly covered by shrubs and trees [5,57]. Other authors report that spontaneous succession tends to fail in stressed or very productive habitats, and is successful in the intermediate stage of the productivity-stress gradient [55].

The spontaneous succession pattern in old fields shows that perennials became established in less than four years and remained for over 50 years. There is an initial stage of decreasing annuals that are replaced by perennial herbs, followed by some decades of perennial grasslands or shrubland development. Species richness fluctuates in the first stage of succession due to the turnover between annuals and perennials; the species number subsequently becomes quite stable. Allochthonous species are common in these old fields, but diminish throughout the succession. The success of the succession is influenced by the species pool in the surrounding areas and by seed dispersal [5,34,58,59,60]. Spontaneous recolonization is very difficult in areas with intensive agriculture, and particularly if soils were enriched intensively before the abandonment with a nutrient such as phosphorus. If not, meadows and perennial grasslands in central European areas can undergo spontaneous recolonization more easily [61]. In deforested lands with grassland cover, native trees can accelerate spontaneous succession. Pioneer species dominate the first stages of succession over non-pioneer species; some studies have observed a higher survival rate for non-pioneer species due to their richer seed bank and competence abilities [54,62]. Some morphological traits (i.e., height) or the number of individuals in a species provide information on vegetation dynamics, such as which species groups were the first invaders, whether there are facilitation or competition processes, or future trends. The colonization speed on south-facing sites may be slower because of insolation, higher temperatures and water stress for plants, while in north-facing areas, woody species can increase their cover faster from one stage to another [56].

Passive and spontaneous regeneration bring us a valuable information on former land uses of present abandoned lands.

## 3. Biotic Interactions and Abiotic Filtering

Biotic limits, which depend directly on biodiversity (i.e., species competition), and abiotic filtering, which depends on the environment, become particularly important when they are exceeded, making it more complicated in farmlands to reach the previous stage. For instance, if a biotic limit is exceeded, the vegetation must be manipulated with a new combination of species to recover. If an abiotic filter is exceeded, the environment needs to be modified; in old fields with a strong biotic and abiotic legacy, there may be a permanent stage of degradation due to poor seed distribution and competition with invasive plants. This phenomenon is more exacerbated in Mediterranean areas because of their more limited soils and changes produced by annual pioneer vegetation to the resources necessary for perennial species [63,64].

The interaction between plants and insects is a good example of biotic interaction. Herbivorous insects can alter the plant community by activating changes in the expression of defense traits [65]. Plants are known to exhibit phenotypic plasticity to both biotic and abiotic conditions in their environments [66]. During succession in specific plant communities, the environmental heterogeneity and the genetic composition of a population can determine its dominant phenotype [65,67].

Studies on biotic and abiotic limits in secondary succession are useful for restoration. The natural vegetation dynamic effectively improves ecosystem development but takes 20 years. This time can be shortened with management of well-conserved areas close to sites requiring restoration (roadsides) in order to improve the plant composition in late-successional stages, when soil loss should be avoided with any biological solution (soil biocrust, etc.) if possible, and soil C and N accumulation should be improved [68].

Rainfall is a direct abiotic filter since it is a resource in itself and is important for recovery. Together with soil type and substrata, it determines which species will develop after abandonment. Rainfall is also influenced by climate change, and as a direct filter, its influence can be unavoidable [69]. Other studies on the abiotic limit in secondary succession focus on pH, which declines after 28 years of abandonment in acidic sandy soils [69].

Biotic interactions and abiotic filters have been included together as ecological rules for dispersal limitation [70,71]. These are defined as the mechanisms or processes that govern the coexistence between species in plant communities. These rules or filters act in scales, from broader to narrower, until the community level [71,72,73]. The local species pool is then determined from the regional pool from the first filter, namely the dispersal limitation, after which environmental limitations (abiotic filters) act on the potential species to colonize these sites, determining the composition. Finally positive and negative species interaction limits the coexistence between species [74,75].

Biotic interactions and abiotic filtering play an important role in changes occurring in a secondary succession. They can reveal how any of these can affect the future stages of a secondary succession.

## 4. Species Pool, Priority Effects, and Species Coexistence

The species pool was defined by Eriksson [76] as the set of species that are potentially capable of coexisting in a certain community. Other further features related to the spatial scale were introduced in the geographical distribution (regional pool), landscape type (local pool) and target community (community pool) [77,78,79,80]. These were used to explain community richness and as an alternative to species coexistence [78]. This definition has been expanded to introduce abiotic, dispersion and co-occurrence concepts to refer to species that are ecologically suited to a particular habitat [50,78,81]. The functional aspects of the species pool throughout the set of trait values have also been taken into account in these types of studies [82,83].

Priority effects were defined as a competitive process in the model proposed by Slatkin [77], which refers to the effect one species can exert on another when becoming established [45,84,85]. This phenomenon should be understood within the specific history of community assembly, and together with probabilistic dispersal, has a greater influence at finer scales [50,86]. Assembly history can influence the structure (richness and composition) and functional properties of ecosystems, and energy flow [29,84,85,87].

There are many studies seeking to understand and clarify how species coexist and the mechanism of species assembly, but also much debate [29,45,73,88,89,90]. The processes involved in diversity patterns can vary at different scales, which makes it difficult to study the rules of assembly that drive the coexistence of species at a local scale. Biogeographical processes and macroevolutionary events and other phylogeographical processes are responsible for diversity patterns at higher scales and for the species regional pool, a source on which other processes will act on local scales [49,71,74,75,79,84,91,92].

Plant-trait based plant ecology or functional plant ecology [53,93] consists of extrapolating plant functional traits from individuals to communities and can be connected to other ecological perspectives such as niche theory, competition, assemblage and ecological filters [90,94]. These plant functional traits are related to the morphological, physiological, behavioral, and phenological features of individuals and species that can affect fitness or persistence over time, and also detect previous selective pressures from biotic or abiotic filters [95,96]. Some features of plant functional traits in secondary succession are described in more detail below.

### 4.1. Plant Functional Traits

Plant functional traits (hereafter PFTs) are morphological, biochemical, physiological, structural, phenological, or behavioral characteristics of organisms that influence their performance or fitness and are widely used to group species. Two common applications of PFTs are: (i) to characterize the responses of plant communities to environmental changes, and (ii) to quantify the influence of community changes on the ecosystem processes [97]. They are used in secondary succession studies to establish differences between abandonment stages or to compare types of vegetation or flora and are more closely related to the functioning of plant communities [82,98].

The choice of the traits studied strongly affects the results, and the combination of traits can also alter the patterns observed. During the transition of stages in the succession, functional changes reflect a general pattern from a ruderal and quick-growing to a more competitive-conservative strategy due to the decreasing availability of light and nutrients during the succession [99]. These results disagree with those of other studies on the relationship between functional divergence and the succession stage, which are negative [100,101], positive [102,103,104], similar [105,106], or without any correlation [107]. This disagreement may be because the authors do not consider the dimension of the traits (individual or group) when detecting different succession mechanisms. Another explanation is that the results should be specific to a determined ecosystem. The interactions between the local environment and the regional climate conditions can also affect the succession patterns [99,108,109].

One interesting trait in secondary succession is the leaf life span (hereafter LLS, see Table 2), which indicates the balance between fast biomass production and efficient nutrient conservation. Low LLS is associated with fast ratios of C fixation and relatively slow growth. If LLS is higher, then nutrients remain for longer in the plant. LLS can vary with the habitat characteristics. Fast-growing species with low LLS tend to occupy nutrient-rich sites, while slow-growing species with high LLS are favored in nutrient-poor areas. Differences in LLS vary widely depending on the abscission period and the time between the end of leaf out and leaf fall [110]. However, there is a codependency between LLS variation and other leaf traits such as Nmass (area-based leaf N), Amass (mass of species), SLA (specific leaf area), RGR (plant relative growth rate), and the growth rate in different ecosystems in spite of the difference in climate, resource availability, leaf form, and seasonal stress [111]. There are also differences between the LLS in young and mature individuals that significantly impact productivity [82].

PFTs can be used as indicators of plants’ colonizing or competitive capacity. Some studies reported results using a temporal gradient of old-field abandonment and comparing it with a site that has not been altered in 40 years. The findings showed how the ratio between photosynthesis and proline content in leaves was positively correlated with species abundance in the initial stage of succession and negatively correlated in the late stages. The germination rate was negatively correlated with species abundance in the early stages and negatively in the late stages of the non-altered site. These results are related to the replacement of pioneer species in the early stages with competitive species in the late stages [54,62].

Plant functional traits are the tools that allow us to see variations among species. However, as we mentioned before results can be different according to the combination of the results.

### 4.2. Dispersal Limitation

This functional trait serves to evaluate how plants become established in a secondary succession (Table 2), and is considered an important process from a stochastic point of view (see “Secondary succession dynamic”). If there are no seeds in the soil, the species may disperse in open lands, establish under environmental conditions and compete with other species. Seeds are accessible to pollinators and other dispersant vectors to be able to reproduce and colonize other sites [159]. Dispersal filters are very important for determining the species pool in the first stages of succession. These early stages are associated with species that can be dispersed at greater distances and with a greater frequency of wind-dispersed species [112,113]. Subsequently, in the late succession stages, seed dispersal is done by animals [114]. However, there is no total consensus on the dispersal traits in plant succession [15,115].

When comparing factors that limit species frequency in grasslands and old fields, the frequencies in the two habitat types were more limited and more closely determined by habitat (biotic and abiotic limitation) than by seed availability (source and dispersal limitation) [116]. In the comparison of forest specialist and corridor species, forest species seem to be well adapted to dispersal-related traits, and colonization is not limited by species dispersal traits. However, the prevalence among forest specialists of long-distance dispersal properties and the flexibility to use many reproductive types suggests that dispersal limitation has been overestimated as a factor, particularly when considering colonization as a long-term process [117].

Studies in old fields from 20 to 28 years have used dispersal traits weighted on abundance and unweighted traits. The traits used were the number of means of dispersal for each species, and classification indices for anemochory and epizoochory. Along the succession the authors verified how the richness and dispersal of generalist species diminishes as the species become more specialized. Species richness and epizoochory ranking are indicators of the transition from the early stages of these old fields. Anemochory does not follow similar patterns in different study areas, depending on the succession context [115]. Ectozoochoric species appear only in the early stages and anemochoric species in the intermediate stages; geochoric species increase in the first 20 years and are maintained throughout the succession [118].

Finally, Makoto and Wilson [119] showed how dispersal limitation is a non-climate-dependent environmental filter that is more important than others (i.e., ecosystem development and biological interactions) [74] in the early stages of succession. This could be understood as a general pattern in many types of biomes such as temperate, boreal, arctic and tropical.

Dispersion is one of the more important stochastic processes, determining the species pool in the first stages of a secondary succession with a higher influence of wind on it, then animals are the main dispersal agents in late successional stages.

## 5. Feedback between Plant and Soils

Nitrogen levels in ecosystems are probably one of the main variables determining the succession rate (Table 2). The patterns of successional changes in nitrogen availability are similar to those of mineralization, both of which increase in primary and decrease in secondary succession [120]. However, it is difficult to see the effect on soils if the abandonment is more recent than 20 years [121].

Studies in old fields, when the age of abandonment is known, reveal that microbial communities and the feedback between plants and soils are drivers of species turnover in succession [122,123,124]. When plants start to grow they generate soils with lower nitrogen concentrations. From this starting point, the inorganic nitrogen immobilized by the microbial community increases and the net rate of mineralization diminishes. This increase is maintained for ten years [125]. Other studies have shown an increase in the soil C/N ratio over time but not in other parameters such as TN (total nitrogen), AP (available phosphorus), pH, BD (soil bulk density), and carbonates [126].

C and N cycles in terrestrial ecosystems are related to plant and microbial activity patterns. The common limitation of N for plant growth and C for microbial growth suggests that these patterns are related [127]. Values of organic C (SOC) and soil N decrease in the first years after abandonment (6 years) as the decomposition rates exceed the net production. These values subsequently start to increase [127,128,160] but may be constant up to 30 years [129]. Exposure also affects these levels but is not homogeneous [130,131,161,162].

Soil organic carbon (SOC) in old fields can be affected by rainfall and is a limiting factor for secondary succession and soil characteristics. In a chronosequence and along a rainfall gradient, SOC concentration increases with abandonment time and precipitation. In arid or drylands such as Mediterranean areas, SOC accumulation in secondary-succession soils is slow, while it is faster in areas of high annual rainfall [132,133]. Soil organic matter (SOM) shows a correlation with pH and N in humid areas because they have a very similar relationship with time while nitrogen is the parameter most closely related to abandonment time in drylands [61].

Another important factor in soil nitrogen concentration is the presence of legume species in the area, which fix atmospheric nitrogen. This is a positive driver in SOC accumulation [134], and has been studied in dry and tropical areas with similar results [135,136]. Another important soil trait is erosion, which is more important than climate change. In mountainous abandoned areas in a temperate climate in northern Europe, land-use changes or the disappearance of plant cover can cause undesirable erosion processes. Secondary succession in mountain areas in grasslands and forests can limit soil erosion, particularly in steeply sloping mountains [137,138,139,140].

Extracellular enzymes and microbial biomass play an important role in biochemical soil cycles, carbon dynamics, and soil function and development [6]. After abandonment, vegetation cover changes and soil organic matter and environmental conditions strongly affect microbial activity recovery. Studies on semiarid cultivated lands after abandonment showed lower enzyme activity than other lands abandoned earlier (12–45 years). After 45 years, soils had not recovered their enzymes, which may be a key finding for restoration [6,141,142].

Soil C and N have a more or less established pattern in a secondary succession, being their use highly recommended, although they can be affected by other abiotic factors, such as the rain.

### The Role of the Soil Seed Bank in Succession

The importance of the seed bank for recovering vegetation after a process of disturbance has been widely highlighted and is recognized as an essential community trait for ecosystem functioning [143,144,145,146,147], providing information on past management practices or degradation levels [148,149,150]. It plays an important role in the dispersion of species and contributes to the conservation of the genetic variability of plants in the ecosystem [149].

Similarities between the soil seed bank and the aboveground vegetation have been extensively analyzed with different results depending on the study, vegetation type, and the disturbance [145,146,147,151,152,153,154,163]. Studies on regeneration processes have shown that the seed bank is more important in early secondary succession than the seed rain. In recently disturbed grasslands or in habitats suffering periodic disturbance there is a higher presence of soil seed bank species and aboveground composition that can have consequences on the future community structure due to the greater opportunity of some species to be dominant, which can greatly influence the seed bank [145,155].

Other studies on seed banks in the short term or by chronosequence have shown a gradual decrease in seeds with greater depth. After perturbation, shallow soils show lower species richness and abundance, although the first succession is constant in the seed bank [145,156,157,158].

Soil seed banks have a recognized role in plant community developing but more studies are needed to establish a more precise pattern.

## 6. Chronosequence

The chronosequence is the set of sites that share the same substrate but differ in regard to the time they were formed. They are appropriate for studying plant succession at different time scales when there is evidence of similar processes, and communities with convergent successional histories, low diversity, fast species turnover, and presence of ecological succession processes [124].

### 6.1. Parameters in a Chronosequence

To prepare a chronosequence it is necessary to have different nearby sites with a similar use (agricultural, forest, crops, pastures, etc.) with evidence of abandonment or changes in use (Figure 1). Abandonment should occur at different times to obtain representative data for a more or less extensive time period. The year of abandonment can be determined with greater accuracy by observing aerial photographs from different years. However, this approach poses certain problems such as the lack of availability of aerial photos for all the years, especially the oldest. The images may have an inadequate resolution and may not be able to be easily dated. Sites are sometimes abandoned for a few years and then cultivated again, so it is essential to check whether the abandonment remains unaltered [23,61,62,164]. There are some preliminary concepts for assessing a chronosequence such as ecological succession, changes in the composition/structure of species over time, and the different types of severity (primary or secondary). The biomass, nutrient availability, and height of the vegetation can indicate an increase (progressive succession) or decrease (regressive succession) [124,165].

Temporal changes in biotic and abiotic aspects of the soil must be observed, including water availability, nutrients, structure, texture, and biota. Disturbances are responsible for the sudden loss of biomass or structure of an ecosystem but create opportunities for establishment by altering the resources or the physical environment, in addition to initiating and modifying succession. Organisms have complex responses that impact biodiversity [2,166,167,168]. The time scale is important, and influences the interpretation of these previous conditions. Succession is normally studied on a time scale that represents 1–10 times the life expectancy of the dominant species [169]. There are multiple potential trajectories in succession, including single or multiple pathways that can be parallel, convergent, divergent, or cyclical [169]. Convergence occurs due to the reduction in the heterogeneity of the species composition. When the successional trajectories are divergent or non-linear, the chronosequence is less useful and requires more sampling than in parallel or convergent cases. Temporal variables are hard to interpret due to the spatial variability of the soil compounds and their influence on the plant and soil communities. The time and severity of the interventions may be unknown, making it more difficult to establish a chronosequence. Conversely, when these interventions are well documented, the details of their timing and severity can help clarify the trajectory and improve their implementation [124].

To determine whether the chronosequence is correct, each of the chosen sites must be resampled after a certain time to check if the predicted sequence occurs. This resampling also provides other data that is not accessible from a normal chronosequence, such as the successional change rate after the intervention. Foster and Tilman [170] measured these rates and tested the hypothesis that they decline over time; they also evaluated the quality of the successional patterns of change in both species richness and abundance, and observed a decrease in the succession rate, which may be due to differences in species growth rate and longevity. They observed that there was no increase in species richness between the sampling dates, perhaps because this period was too short (14 years). Species richness may increase in a non-linear way, or this may not be the most appropriate method to predict changes in species richness. Finally, resampling confirmed that the succession rate decreases over time [170].

Disturbation is one of the main drivers to face when starting a chronosequence. It is not always possible to know its origin, being necessary to control as much variables as possible.

### 6.2. Case Studies in Mediterranean Areas

The comparison of the soil function in a forest stand with natural afforestation (secondary succession) in old fields showed a non-linear soil improvement in the old fields, whereas this improvement was linear in the forest stand. After 40 years, the soil status in the forest stand is comparable to semi-natural vegetation, with typical resource islands (soil quality indicators patching). Finally, it takes longer to achieve similar soil recovery in old fields than in afforestation [171].

Another chronosequence study in abandoned crops at five and fifteen years with natural vegetation revealed an increase in diversity and richness of all functional groups in the fields that had been abandoned for a longer time compared to more recently abandoned fields. Diversity was also affected by mechanization, and by the fact that species with long-lived seeds cannot recover in the early stage of succession. As in other secondary succession cases, the recovery of annuals occurred only in the early stages, while the transition to perennials took place after 15 years of abandonment. Total nitrogen increased in the most recently abandoned fields and the percentage of sand was higher in recent old fields, as cultivation techniques produce a gradual degradation of the upper soil layers that is more severe than erosion. A negative correlation was also found between richness and diversity in all functional groups with silt. This is because silt changes the soil pH and reduces the availability of micronutrients, zinc, manganese, and P [98].

Like many other croplands, vineyards have also been abandoned. This is due to the policies of the European Union and France, which promoted changes in the quality of wines and cultivation methods, while subsidizing the displacement of non-productive vineyards, which encouraged abandonment. Where these changes were not implemented, the vineyards were abandoned [22]. These abandoned vineyards are used to develop a chronosequence. Recent decades have seen a decrease in cultivated hectares in some countries such as Spain due to social factors including rural flight, and environmental limits such as the depth and slope of the soils or the availability of water [164]. A study on soils in a chronosequence of 59 years showed how total organic carbon (TOC) increased in the succession up to 25 years before stabilizing. Enzyme activity also increased throughout the succession, with a greater change in the first decade. There is also an increase in plant density between 7 and 11 years of abandonment [126,172].

In Mediterranean climates there are many vineyards abandoned; it would be necessary to develop more successional studies on them.

## 7. Climate Change and Secondary Succession

The data indicate that the anomalous climate of the past half century is affecting the physiology, distribution and phenology of species. Although natural climate change and other non-climatic factors may be responsible for these alterations, human-induced climate and atmospheric change is now the more consistent explanation [3,173,174,175,176].

The study by Thuiller et al. [177] took data on the distribution of 2294 species (accounting for 20% of the total European flora) from samples collected between 1972 and 1996, which are sufficiently representative as they include most life forms and phytogeographic patterns in Europe. The results they obtained making predictions for the future in seven different scenarios depend on the existence or not of universal migration. In the case of non-migration, more than half of the species may be vulnerable or threatened with extinction by 2080. In contrast, climate change impacts are more negligible if there is universal migration due to the possibility of species moving across the terrain, as when a species is restricted to a few places, local catastrophic events (droughts), or an increase in the transformation of the land by humans can cause its extinction.

Focusing more on Spain, they obtained two different predictions for northern and southern zones. In the center-north there was an excess of loss of species as these were habitats with little tolerance, and were marginal for most species. In the southern zone, however, there is not loss of species. This is due to the dry warm summers that enable these species to successfully tolerate heat and drought, making them potentially well adapted to future changes [177].

The impact of CO_2_ on global warming has led to a growing interest in reducing emissions and increasing their sequestration in the soil. This a good option, since abandoned lands can be a low-cost strategy to sequester C and mitigate anthropogenic CO_2_ emissions [128].

Carrying out a long-term chronosequence can allow us not only to compare different variables across time but also to observe changes due to warming and their direct consequences on secondary succession plant communities.

## 8. Conclusions

The presence of old fields today allows us to investigate them and learn about their legacy in ancient times. We can determine how these lands recover after they have ceased to be used for growing crops. By doing this with a temporal gradient of sampling areas, we can estimate the time it takes for land to recover to its usual state after being manipulated. In this review, we have seen all the factors that must be considered when carrying out a chronosequence. It is not always possible to guarantee that all these factors can be correctly identified. If it is done correctly, we will obtain a large amount of data providing information on how the secondary succession has occurred or is occurring, which allows us to develop hypotheses. All these data can also be helpful to develop further works in abandoned lands, for instance, a restoration. The results obtained in the chronosequence can guide us in determining what type of use is most recommended.

Another important aspect is the relationship of all of the above with climate change. The availability of a chronosequence from fields abandoned 30 years ago to fields abandoned today allows us to quantify the effects of climate change on vegetation, which have become more significant in recent years.

However, more studies are needed that combine measurements of several PFTs so that the hypotheses that arise are more exact, such as soil analyses and measurements of chlorophyll and proline in plants [178]. Despite all this, the preparation of chronosequences for secondary succession has limitations due to the wide time scales with which it works. Since it is difficult to maintain control over such wide time ranges, it is necessary to develop tools or methods with which to prepare a chronosequence in the most exact way possible and to determine with the utmost accuracy what has occurred in each terrain in previous years.

## Figures and Tables

**Figure 1 plants-10-00990-f001:**
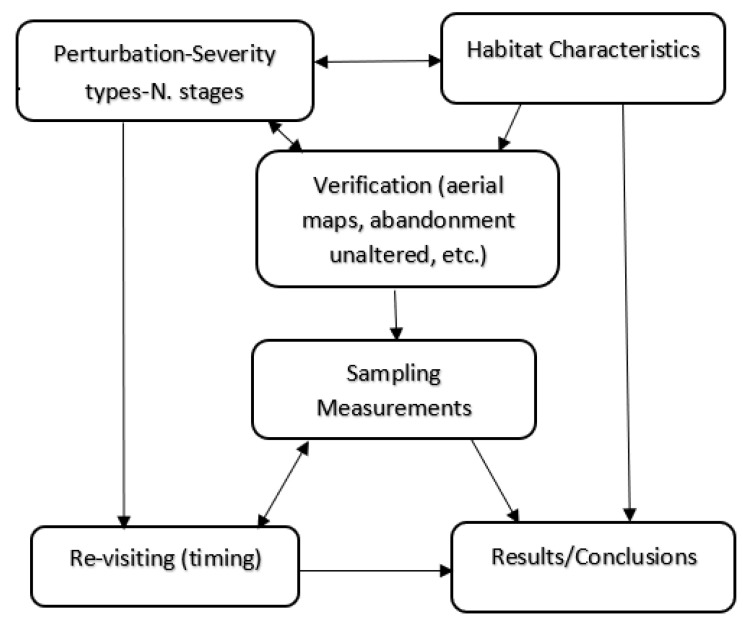
Road map of a chronosequence. Numbers indicate the different steps to follow when starting a succession study. Connections between steps are made by arrows; double arrows indicate a feed-back process that in the case of sampling and revisiting means short or medium term sampling (see text).

**Table 1 plants-10-00990-t001:** Summary of main themes, important aspects, drivers and outcomes enclosed in the revised literature for secondary succession. Abbreviations are: LLS (Leaf Life span); Nmass (area-based leaf N content); Amass (species mass); SLA (specific leaf area); RGR (plant Relative Growth Rate); C (carbon); N (nitrogen); AP (available Phosphorus); SOC (soil organic carbon).

Main Themes	Aspects	Drivers	Outcomes
Dynamic of a secondary succession	Deterministic.	Abiotic resources availability. Biotic limits.	Succession pattern. Velocity of succession.
	Stochastic.	Species pool. Functional changes. Propagule dispersion.	Plant community structure. Consistency of secondary succession.
Biotic and abiotic limits	Biotic interactions.	Competition/facilitation. Plant–animal interactions.	Phenotypic plasticity. Restoration.
	Abiotic filters.	Rainfall and climatic parameters. Soil characteristics.	Phenotypic plasticity. Restoration.
Species pool, priority effects and species coexistence	Regional and local pool/ assembly rules.	Richness, composition, functional properties, energy flow.	Diversity patterns. Local scale. Plant community structure.
	Plant Functional Types (PFTs).	LLS, Nmass, Amass, SLA, RGR.	Biomass productivity. Efficient conservation.
	Dispersal limitation.	Habitat type. Seed availability.	Long-dispersal. Short-dispersal.
Feedback between plant and soils	Soil characteristics.	Nitrogen concentration, C/N ratio, AP, SOC.	Atmospheric nitrogen fixation.
	Soil seed bank.	Ecosystem functioning, degradation levels, management practices.	Vegetation recovering. Community structure.
Chronosequence	Parameters in a chronosequence.	Disturbance type. Soil spatial variability. Resampling.	Structure of species over time.
Climate change		Vulnerability. Migration.	Local events (drought). Species extinctions.

**Table 2 plants-10-00990-t002:** Important drivers in a secondary succession.

Driver	Effects	References
Leaf life span (LLS)	Indicates the balance between rapid biomass production and efficient nutrient conservation	[110]
Dispersal traits	Evaluates the stabilization of plants after a secondary succession	[112,113,114,115,116,117,118,119]
C, N, SOC, TOC	Analyzes the presence of these organic components in secondary succession soils	[61,120,121,122,123,124,125,126,127,128,129,130,131,132,133,134,135,136,137,138,139,140]
Enzyme activity of the soil microbiota		[6,141,142]
Seed banks	Analyzes the presence of different seeds after secondary succession and their contribution to plant development	[143,144,145,146,147,148,149,150,151,152,153,154,155,156,157,158]

## Data Availability

Not applicable.

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
