# Peer review of "Impacts of Land-Use Changes on Vegetation and Ecosystem Functioning: Old-Field Secondary Succession"

_plants, 2021, doi:10.3390/plants10050990_

Round 1

Reviewer 1 Report

This review has the potential to be published in plants. But, I have some comments.

Restructure abstract.

Every section lacks the authors summary...e.g. every section except the introduction should have author conclusions, not just complied information with references. 

Elaborate conclusion.

Break larger sentences into short

Several language mistakes exist, have a careful checkup with an English Language native.

Figure 1 is confusing.

Author Response

Dear sir/madam,

We are sending the revision of our paper entitled: Impacts of land-use changes on vegetation and ecosystem functioning: old-field secondary succession, according to the reviewer’s reports. Specifically we have answered any of the items that are listed below. Hopefully the new version is suitable for publishing in Plants.

Yours sincerely

Javier Pérez-Hernández & Rosario G. Gavilán

This review has the potential to be published in plants. But, I have some comments.

POINT 1Restructure abstract.

RESPONSE TO POINT 1:Done; it has been modified to make it more informative.

POINT 2: Every section lacks the authors summary...e.g. every section except the introduction should have author conclusions, not just complied information with references. 

RESPONSE TO POINT 2: Done; we have added at the end of every chapter some conclusions.

POINT 3:Elaborate conclusion.

RESPONSE TO POINT 3:We do not understand what the referee means, if this sentence is connected to the former paragraph we did it. In any case, we have checked the conclusions chapter and made changes on it.

POINT 4: Break larger sentences into short.Several language mistakes exist, have a careful checkup with an English Language native.

RESPONSE TO POINT 4: The English have been corrected by an English speaker and long sentences are allowed in English as in other languages. We have added an acknowledgements at the end of the paper to thank the English correction.

POINT 5: Figure 1 is confusing.

RESONSE TO POINT 5: We have added explanations to the caption of the figure 1

Reviewer 2 Report

Review of ms: Impacts of land-use changes on vegetation and ecosystem functioning: old-field secondary succession by Perez-Hermandez and Gavilan for Plants.

This manuscript provides a framework to summarize theoretical and data-based studies of old-field succession. I found it useful at integrating biotic and abiotic factors influencing successions. It also usefully covers both American and European studies. The writing in general was easy to follow. I occasionally did find the writing to be too compact and jargon-filled. I have relatively few specific suggestions.

1) line 57. Delete “some of which were limited to particular areas of France” since the following sentence includes studies from both France and England.

2) line 59. Delete “and”.

3) lines 74-79. Models i, ii, iii, and iv overlapped to me.

4) lines 99-100. Are you saying that light is more important than nutrients for woody plants in succession? I believe it but am not certain I interpreted your wording accurately.

5) Table 1. I am unclear what is meant by “Amass, mass of species”.

6) Table 1. What is meant by “success of secondary succession”?

7) Section “secondary succession dynamic: deterministic or stochastic processes?” Half way through this section the contrast seemed to change from deterministic/stochastic to passive/spontaneous regeneration. Define the latter terms and show how they relate to the former.

8) Table 2. I could not find this table referenced in the text. If I am wrong, I apologize.

9) lines 464-473. Expand the description of the study. By “plants” do you mean “species”? Did the study talk of recent migration or did it make predictions for the future?

10) line 477. Is a “negative loss” a gain?

11) lines 493-494. Give an example of “another use”.

Author Response

Dear sir/madam,

We are sending the revision of our paper entitled: Impacts of land-use changes on vegetation and ecosystem functioning: old-field secondary succession, according to the reviewer’s reports. Specifically we have answered any of the items that are listed below. Hopefully the new version is suitable for publishing in Plants.

Yours sincerely

Javier Pérez-Hernández & Rosario G. Gavilán

POINT 1) line 57. Delete “some of which were limited to particular areas of France” since the following sentence includes studies from both France and England.

RESPONSE TO POINT 1: Done

POINT 2) line 59. Delete “and”.

RESPONSE TO POINT 2: Done

POINT 3) lines 74-79. Models i, ii, iii, and iv overlapped to me.

RESPONSE TO POINT 3:The use of lower case roman numbers is a quite common and elegant way to make a numeration in a text. It is commonly use in scientific books, reports, papers, etc. For us, it is not an objection to change them but we would like editors and reviewer indicate us how to proceed.

POINT 4) lines 99-100. Are you saying that light is more important than nutrients for woody plants in succession? I believe it but am not certain I interpreted your wording accurately

RESPONSE TO POINT 4:. No, nutrients and particularly N is a major force for woody plants. We have added a sentence to clarify it.

POINT 5) Table 1. I am unclear what is meant by “Amass, mass of species”.

RESPONSE TO POINT 5:Done. We have corrected it and change by ‘species mass’.

POINT 6) Table 1. What is meant by “success of secondary succession”?

RESPONSE TO POINT 6: Done. We have change the term ‘success’ by ‘consistency’, more appropriate.

POINT 7) Section “secondary succession dynamic: deterministic or stochastic processes?” Half way through this section the contrast seemed to change from deterministic/stochastic to passive/spontaneous regeneration. Define the latter terms and show how they relate to the former.

RESPONSE TO POINT 7: Done. We have made changes in the text to make it more clear.

POINT 8) Table 2. I could not find this table referenced in the text. If I am wrong, I apologize.

RESPONSE TO POINT 8: Yes, we have not included it in the text, now it is, thank you :)

POINT 9) lines 464-473. Expand the description of the study. By “plants” do you mean “species”? Did the study talk of recent migration or did it make predictions for the future?

RESPONSE TO POINT 9: Yes and yes. We have make changes and included the term species instead of plants and make predictions for the future in seven climate change scenarios.

POINT 10) line 477. Is a “negative loss” a gain?

RESPONSE TO POINT 10: Done. It is not really a negative loss but just no loss at all! Thank you for the feedback! We have made changes in the text.

POINT 11) lines 493-494. Give an example of “another use”.

RESPONSE TO POINT 11:Done. We have made changes in the text more according with the suggestion.

Reviewer 3 Report

This Review deals with impacts of land-use changes on vegetation. The topic is interesting. The Review is well written, quite complete and presenting a good bibliographic background. The English is ok. I suggest acceptance after a few minor revisions:

Keywords: Do not repeat words already included in the Title, i.e. delete “secondary succession” and “old-fields”

Keywords: Why not alphabetic order?

Keywords: “soil seed bank” instead of “soild seed bank”

Line 49: “km2” instead of “km2”

Lines 58-59: something is missing. Maybe a comma? “in France [13-14] French landscape since prehistory”

Lines 75-79: Use the same font as in the whole text

Line 77: “[25-29]; ii)” instead of “[25-29] ii)”

Lines 92-93: Use the same font as in the whole text (also in other parts of the manuscript, e.g. lines 118-126, 224-225, etc.)

Line 129: Delete a point: “55]. For instance”

Line 254: “patterns [99,108,109].” Instead of “patterns [99], [108,109].”

Table 2: in “References” column insert the numbers of the references, as they can be found in the reference list at the end of the manuscript. If not all the references reported in Table 2 are included in the reference list, add them to the reference list at the end of the manuscript.

Author Response

Dear sir/madam,

We are sending the revision of our paper entitled: Impacts of land-use changes on vegetation and ecosystem functioning: old-field secondary succession, according to the reviewer’s reports. Specifically we have answered any of the items that are listed below. Hopefully the new version is suitable for publishing in Plants.

Yours sincerely

Javier Pérez-Hernández & Rosario G. Gavilán

This Review deals with impacts of land-use changes on vegetation. The topic is interesting. The Review is well written, quite complete and presenting a good bibliographic background. The English is ok. I suggest acceptance after a few minor revisions:

POINT 1. Keywords: Do not repeat words already included in the Title, i.e. delete “secondary succession” and “old-fields”.

RESPONSE TO PINT 1: Done.

POINT 2. Keywords: Why not alphabetic order?

RESPONSE TO POINT 2: Done.

POINT 3. Keywords: “soil seed bank” instead of “soild seed bank”

RESPONSE TO POINT 3: Done.

POINT 4. Line 49: “km2” instead of “km2”

RESPONSE TO PINT 4: Done.

POINT 5: Lines 58-59: something is missing. Maybe a comma? “in France [13-14] French landscape since prehistory”

RESPONSE TO POINT 5: Done.

POINT 6. Lines 75-79: Use the same font as in the whole text

RESPONSE TO POINT 6: Done.

POINT 7. Line 77: “[25-29]; ii)” instead of “[25-29] ii)”

RESPONSE TO POINT 7: Done.

POINT 8. Lines 92-93: Use the same font as in the whole text (also in other parts of the manuscript, e.g. lines 118-126, 224-225, etc.)

RESPONSE TO POINT 8: Done.

POINT 9. Line 129: Delete a point: “55]. For instance”

RESPONSE TO POINT 9: Done.

POINT 10Line 254: “patterns [99,108,109].” Instead of “patterns [99], [108,109].” RESPONSE TO POINT 10: Done.

POINT 11.Table 2: in “References” column insert the numbers of the references, as they can be found in the reference list at the end of the manuscript. If not all the references reported in Table 2 are included in the reference list, add them to the reference list at the end of the manuscript.

RESPONSE TO POINT 11: Done.
